# Differential Adaptation Has Resulted in Aggressiveness Variation of *Calonectria pseudonaviculata* on Hosts *Buxus*, *Pachysandra*, and *Sarcococca*

**DOI:** 10.3390/jof9020181

**Published:** 2023-01-29

**Authors:** Ping Kong, Margery L. Daughtrey, Chuanxue Hong

**Affiliations:** 1Hampton Roads Agricultural Research and Extension Center, Virginia Tech, Virginia Beach, VA 23455, USA; 2Long Island Horticultural Research and Extension Center, Cornell University, Riverhead, NY 11901, USA

**Keywords:** plant–pathogen interaction, boxwood (*Buxus*), pachysandra, sweet box (*Sarcococca*), *Calonectria pseudonaviculata* (*Cps*), host adaptation, aggressiveness, infectibility, lesion size, sporulation

## Abstract

*Calonectria pseudonaviculata* (*Cps*) infects *Buxus* (boxwood), *Pachysandra* (pachysandra), and *Sarcococca* spp. (sweet box); yet, how it adapts to its hosts has been unclear. Here, we performed serial passage experiments with the three hosts and measured *Cps* changes in three aggressiveness components: infectibility, lesion size, and conidial production. The detached leaves of individual hosts were inoculated with isolates (P0) from the originating host, followed by nine serial inoculations of new leaves of the same host with conidia from the infected leaves of the previous inoculation. All boxwood isolates maintained their capability of infection and lesion expansion through the 10 passages, whereas most non-boxwood isolates lost these abilities during the passages. Isolates from plants of origin (*-P0) and their descendants isolated from passages 5 (*-P5) and 10 (*-P10) were used to evaluate aggressiveness changes on all three hosts with cross-inoculation. While post-passage boxwood isolates gave enlarged lesions on pachysandra, sweet box P5 and pachysandra P10 isolates showed reduced aggressiveness on all hosts. *Cps* appears to be most adapted to boxwood and less adapted to sweet box and pachysandra. These results suggest speciation of *Cps*, with its coevolutionary pace with the hosts the fastest with boxwood, intermediate with sweet box, and the slowest with pachysandra.

## 1. Introduction

Boxwood (*Buxus*), pachysandra (*Pachysandra*), and sweet box (*Sarcococca*) are three genera of *Buxaceae* that include important evergreen plants for gardens and landscapes. However, these plants are susceptible to *Calonectria pseudonaviculata* (*Cps*), a fungal pathogen in the family *Nectriaceae* of the Ascomycota, which mainly relies on its asexually reproducing stage for infection [1,2,3]. *Cps* causes the devastating disease known as boxwood blight. Boxwood blight was first found in Europe in the mid-1990s and has spread to many countries of the world, where it damages both cultivated plants and native boxwood ecosystems [4,5]. The first decades of study on this new disease were focused on two of the three important components in the disease triangle, plant susceptibility, and weather conduciveness [1,2,3,4,5,6,7,8]. A study of the third component, the pathogen, and especially its aggressiveness and adaptation to different host plants, was still lacking.

Aggressiveness in plant pathology refers to the quantitative variation of pathogenicity on susceptible hosts, which is composed of multiple phenotypic traits [9]. Infection efficiency, lesion size, and spore production rate associated with host–pathogen interaction are the most commonly considered traits although aggressiveness can be assessed by evaluating additional quantitative phenotypic traits of the pathogen that are directly linked to its fitness [10]. The term “aggressiveness” sometimes is used interchangeably with “fitness” and “virulence”. Fitness has been defined as the ability of an organism to propagate and evolve within a given environment [11], while virulence is described as the quantity of damage induced by the pathogen on its host [12]. For example, the average number of secondary lesions produced from a single lesion on its host has been used to measure fitness [9]. For a filamentous fungal pathogen, fitness is often measured by observing the mycelial growth rate and the number of spores produced [13]. Virulence, on the other hand, is measured in units of host symptoms and mortality. The term aggressiveness combines both notions and links them to evolutionary epidemiology [14].

The aggressiveness of *Cps* has not been specifically studied although some of the aggressiveness components have been investigated in pathogen biology or pathogen–host interaction research. In those studies, *Cps* production of conidia was shown to vary with host genus (*Buxus*, *Pachysandra*, or *Sarcococca*) [15] as well as with boxwood cultivar [7,16]. When hosts of different genera were compared, in addition to sporulation, variations in infection rate and lesion size were also host-dependent [15]. The presence of phenotypic variation in aggressiveness is considered a key factor for pathogen adaptation [10]. Thus, the differential aggressiveness presented by the isolates from different genera of Buxaceae and boxwood cultivars suggests that *Cps* may undergo speciation according to selection pressures from hosts.

Host-mediated speciation has been documented for many fungal plant pathogens and is one of the main routes for the emergence of new plant diseases [10,17,18,19,20]. Some life-history traits including host-imposed disruptive selection, low gene diversity, production of a large number of spores, and frequent asexual reproduction can facilitate the rapid ecological divergence of pathogens [21]. *Cps* has a life cycle starting from penetration of the host plant with germ tubes issuing from germinated spores, followed by hyphal or mycelial growth, asexual conidial production, and formation of microsclerotia; yet there is no clear event of sexual reproduction [16,22,23], perhaps due to lack of different mating types [24]. Furthermore, natural populations of *Cps* have limited genetic diversity, which may be associated with limited opportunities for mating combined with predominately clonal asexual reproduction [25,26]. Given these traits and the known differential aggressiveness of *Cps* in different genera, eventual speciation due to the selection pressure of different hosts is expected although how long it will take for speciation to occur remains unknown.

In this study, we conducted serial-passage experiments to further understand the differential aggressiveness of *Cps* and its impact on boxwood blight disease. Serial-passage experiments are a form of experimental evolution used to study pathogen adaptation and co-evolution between host and pathogen. With this approach, rapid evolution has been detected in a few phytopathosystems [27,28,29,30,31], but no such studies have been performed to date with *Calonectria* species. Here, we quantified changes in three aggressiveness components (infectivity, lesion size, and conidial production) of *Cps* in its originating host, namely boxwood, pachysandra, or sweet box, after each of the ten serial passages. We wished to determine whether the differential host aggressiveness observed previously [15] might be apparent during passages and, if so, to see which isolates and their aggressiveness components might be more likely to evolve in response to host-selection pressures. Furthermore, we performed a cross-inoculation experiment using the cultures of three original isolates with different host origin and their mid-passage and end-passage descendants to determine how the passage-associated changes might affect the ability of *Cps* to infect other host plants, develop lesions, and sporulate on those hosts.

## 2. Materials and Methods

### 2.1. Pathogen Isolates

Three sets of *Calonectria pseudonaviculata* (*Cps*) isolates [15] were evaluated in this study. Each set included three isolates originating from boxwood (*Buxus*), pachysandra (*Pachysandra*), or sweet box (*Sarcococca*) plants within the same garden [3,32] (Table 1). During the experiments, all the isolates were subcultured and maintained on potato dextrose agar (PDA, Sigma-Aldrich, St. Louis, MO, USA) at 25 °C after their retrieval from storage on PDA slants covered with mineral oil at 20 °C.

### 2.2. Plant Materials for Inoculation

Boxwood (*Buxus sempervirens* ‘Suffruticosa’), pachysandra (*Pachysandra terminalis*), and sweet box (*Sarcococca hookeriana* var. *humilis*) were used. For inoculation, unblemished leaves of similar size and age were collected from healthy specimens of each of these host plants and used after surface sterilization with 10% Clorox^®^ Performance Bleach with CLOROMAX^®^—Concentrated Formula (Clorox Co., Oakland, CA, USA) for 1 min, rinsing five times with distilled water (dH_2_O) and then blot-drying with paper towels. The clean, detached leaves, with the abaxial surface facing up, were placed in rows onto polypropylene plastic mesh (Industrialnetting.com, Minneapolis, MN, USA) over wet paper towels in a clear plastic container (31 × 23 × 10 cm, Pioneer Plastics Inc., Eagan, MN, USA) and at this point were ready for inoculation.

### 2.3. Serial Passage Experiments and Inoculum Preparation

The scheme of the serial passage experiments is shown in Figure 1a. In total, 10 serial passages were carried out. For the first passage (I), the conidial inocula were prepared from fresh potato dextrose broth (PDB) cultures of original isolates (P0) (Table 1) from storage, as described previously [32]. Inocula for passages II to X were prepared with conidia from infected leaves of the previous passages I to IX. To collect conidia from the leaves, the infected leaves of boxwood or 1.2 cm^2^ lesion disks of pachysandra or sweet box leaves were excised and vortexed in 0.005% Tween 20 for 5 min at 10 days post inoculation (dpi). The volume of the Tween solution for leaf conidial suspension was 1 mL per leaf. Six and ten detached leaves or lesion disks were used in the preliminary and formal experiments, respectively.

For passage I, a 20 μL drop of the P0 suspension containing approximately 50 conidia was placed in the center of each leaf. For passages II to X, the inoculation was performed as with P0 except that a drop of spore suspension containing approximately 25 conidia from the infected leaves or lesion disks of the previous passage was used. The inoculated leaves were placed in lidded plastic containers at 23 °C with a 9 h day/15 h night cycle under natural and supplementary fluorescent lights at 760 lux for 4 weeks. In formal serial passage experiments, when an inoculation with P0 or its descendants failed at any passage earlier than five, the experiment was restarted to confirm the inability of the P0 isolates or their descendants to infect.

### 2.4. Cps Descendant Isolation during the Passage

Isolation of P0 descendants was conducted during the formal serial passage experiments to obtain P1–P10 isolates for the post-passage experiments. When collecting conidia from infected leaves or lesion disks at 10 dpi for passage inocula, a 10 µL droplet of sterilized distilled water (SDW) was placed onto a lesion of the infected leaf and pipetted up and down to suspend the conidia. To obtain pure cultures of the descendants, five 1 µL droplets of the conidia suspension were placed on a PDA plate and cultured at 23 °C for a week, followed by transferring hyphal tips from the margins of clean colonies to new PDA slants.

### 2.5. Comparative Study with Isolates of Different Passages

The comparative study was performed after the serial passage experiments using three original isolates from boxwood, pachysandra, and sweet box (*-P0) that went through 10 passages and their descendant isolates generated from passages V (*-P5) and X (*-P10) (Table 1, Figure 1b). The experiment was arranged according to the hosts. For each host, a total of 90 detached leaves were divided among three moist chambers, which each accommodated three isolates. Inoculation was performed in the following order: first isolate origin (boxwood, pachysandra, and sweet box) and then passage number (P5 and P10), using conidial suspensions from cultures as described for passage I in the serial passage experiment. Three independent experiments were conducted.

### 2.6. Measurement of Aggressiveness Traits

Three aggressiveness traits, namely infection rate, lesion size, and conidial production, were measured in this study. Only infected leaves were counted at each passage during the preliminary serial passage experiment, but lesion size and conidial production were also recorded during the formal passage experiment and comparative study. Specifically, the number of infected leaves out of a total of 10 inoculated leaves was counted at 14 dpi to determine infection rate (%). The lesion size (cm^2^) of each infected leaf was estimated at 28 dpi by multiplying the percentage of infected leaf area by the average leaf size of 1.72 cm^2^ for boxwood, 9.98 cm^2^ for pachysandra, and 8.05 cm^2^ for sweet box. Production of conidia was measured at 10 dpi by suspending conidia from 10 infected boxwood leaves or 10 pachysandra or sweet box lesion disks in 10 mL Tween solution, as described in the serial passage experiment, and counting the conidia per ml with a hemocytometer under a light microscope. For each suspension, six independent counts were taken. To obtain conidia/cm^2^, the average count of conidia was divided by the average lesion area used for the suspension, which was 1.72 cm^2^ for boxwood and 1.2 cm^2^ for pachysandra and sweet box.

### 2.7. Data Analyses

The serial passage experiment data were analyzed for variance using the Data Analysis function in Microsoft Excel 2010. The data of infected leaves, lesion size, and conidial production by original isolates and their descendants were analyzed with Proc ANOVA of Statistical Analysis Software (SAS Institute, Cary, NC, USA). Data from individual measurements were subjected to a homogeneity test, then pooled as appropriate. The means were separated by the least significant difference test at *p* = 0.05.

## 3. Results

### 3.1. Infectivity Variation on Three Original Hosts during the Serial Passage

In the preliminary serial passage experiments, all three boxwood isolates completed 10 passages with a consistent 100% infection rate on the inoculated leaves at each passage, while only one pachysandra and one sweet box isolate completed all 10 passages. The same results were observed in the formal experiments. All three boxwood isolates infected all test boxwood leaves by 10 dpi at each passage and completed 10 passages (Figure 2a), but only one of the three pachysandra or sweet box isolates completed 10 passages (Figure 2b,c).

The host impact on the infection rate was significant, and the impact of isolates was also significant for isolates of pachysandra and sweet box origin (Table 2). The infection rates varied from passage to passage during passages (Figure 2). Specifically, the isolates that did not complete 10 passages had a sharp infection reduction before or at the 4th passage, while those that survived through 10 passages had a consistently stronger infection. The results indicate that interactions of *Cps* with boxwood were stable, whereas those with other hosts showed poor adaptation to these plants.

### 3.2. Lesion Development Variation from Original Hosts during the Serial Passage

A similar pattern to that seen with infection was observed for the isolates regarding lesion development (Figure 3). Boxwood isolates resulted in lesions on boxwood that expanded so quickly that all inoculated leaves were completely blighted by 10 dpi, and there was no difference in lesion size among the three isolates of boxwood origin (Table 2, Figure 3a). In contrast, pachysandra and sweet box isolates caused different lesion sizes over the course of the passages. Unlike boxwood isolates that caused infection of the whole leaf area, these isolates resulted in necrosis in a small leaf spot. However, differences in lesion size were not significant between these two hosts although they were significant among the isolates of pachysandra and sweet box origin (Table 2). Regarding pachysandra and sweet box isolates that did not complete 10 passages, i.e., ph2, ph11, sb1, and sb6, they caused smaller-sized lesions as passage numbers increased and no lesions after passage IV. In contrast, the two isolates that went through the entire passage series, i.e., ph12 and sb3, exhibited dramatic changes in lesion size during the passages (Figure 3b,c).

### 3.3. Variation in Conidia Production from Original Hosts during the Serial Passage

Of the three aggressiveness components evaluated in the passage, production of conidia was the most affected during the passage series. Sporulation variation was even found among boxwood isolates that had little variation in infection rate and lesion size (Table 2, Figure 4). However, boxwood isolates generally produced more conidia, and the yield was more consistent through the passages compared to that of pachysandra or sweet box isolates. The sporulation reduction in boxwood isolates was not notable until passage VIII, in contrast to sporulation of non-boxwood isolates that showed reduction as early as the 2nd passage (Figure 4b,c). For example, pachysandra isolates ph2-0 and ph11-0 or sweet box isolates sb1-0 and sb6-0, neither of which completed 10 passages, produced fewer conidia than boxwood isolates even at very early passages (Figure 4b,c).

### 3.4. Responses of Original Host and Other Hosts to C. Pseudonaviculata after Passage

To determine whether the changes observed in the passage experiments might be retained in the isolates and how they might affect other hosts, isolates bw1, ph12, and sb3 (*-P0) that completed 10 passages in the serial-passage experiment and their descendant isolates P5 and P10 from passage V and X were evaluated against all three hosts (Figure 1b). The comparative study with these isolates showed that passage, isolates, and hosts as well as experiments all affected the aggressiveness of *Cps* although the least impact was from experiments (Table 3). Interactions of passage with isolate for all the components and with host for production of conidia were also identified, indicating the impact of passage on isolate aggressiveness and host difference for *Cps* sporulation. An interaction also was observed between passage and experiment for infection, indicating that infectivity was the most variable component in aggressiveness affected by passage.

The passage effect on *Cps* aggressiveness was dependent on originating host and number of passages on the same plant. The infection rates of boxwood bw1-P5 and -P10 remained unchanged on all the hosts (Figure 5a). However, bw1-P5 had reduced conidial production on all three hosts (Figure 5c). Likewise, larger lesions were observed on pachysandra with bw1-P5 and -P10 isolates compared to the bw1-P0 isolate (Figure 5b).

Both pachysandra (ph12-P10) and sweet box (sb3-P5) isolates dropped their infection rate on respective originating hosts. Similar drops were also noted in both descendant isolates on the non-originating hosts (Figure 5a) as well as in two other aggressiveness traits: lesion size (Figure 5b) and production of conidia (Figure 5c).

## 4. Discussion

Although *Cps* is known to infect boxwood and its close relatives such as pachysandra and sweet box [5], its adaptation to these hosts has not been evaluated until this study. Through investigation of infection rate, lesion size, and conidial production in serial passage and post passage experiments, this study demonstrated *Cps* aggressiveness variations and potential speciation resulting from a differential adaptation that was the most advanced in boxwood, the least in pachysandra, and intermediate in sweet box. The observed variations in adaptation help us to understand the coevolution of the pathogen with its various hosts; this information is critical for the appropriate selection and utilization of resistant plants for the management of boxwood blight disease.

Adaptive evolution of *Cps* can occur within a season, as shown in this study through passages, or by continuous infection of a specific host, thus changing the course of blight occurrence and development. Boxwood isolates that had smooth and short infection cycles (12.9 days, Table 2) on their originating host are well adapted to this plant. Notably, both bw1-P5 and -P10 isolates consistently caused a significant lesion expansion on pachysandra and sweet box, suggesting improved virulence on other hosts with adaptive evolution. In contrast, non-boxwood isolates are poorly adapted, as indicated by bumpy and longer infection cycles (15.8 days, Table 2) and unstable interaction with pachysandra and sweet box. Obviously, isolates with shorter infection cycles and/or with greater virulence are more harmful. Such isolates or populations may complete up to 28 infection cycles within a year under mild climates. Using other genera of Buxaceae or resistant boxwood cultivars instead of susceptible ones will slow disease development and reduce infection cycles in an infested garden.

Speciation has been reported in clonal fungal pathogens with very low genetic diversity [33], which has been attributed to the occurrence of rapid evolution in these pathogens, providing their relatively small population of descendants with the best possible genetic and environmental background [34,35]. *Cps*, with its very low genetic diversity [25,26,36,37], may be a new addition to this pathogen group although molecular evidence is warranted to ascertain such a hypothesis. In addition to the evidence generated in this study, the sequential order by which this disease spread within the garden where all boxwood, sweet box, and pachysandra isolates evaluated here were collected supports such speculation. All the isolates used in this study were collected within one year of the onset of an epidemic initiated from accidental introduction of infected English boxwood plants [3,32]. In this landscape planting, disease symptoms were noted first on boxwood, then sweet box, and finally pachysandra. These observations suggest that by the time of original isolate collection, *Cps* populations might have undergone several passages on their respective hosts, with a likelihood that more passages occurred on boxwood than on sweet box or pachysandra; these latter hosts probably were infected by inoculum produced on boxwood plants in that garden site.

The existence of phenotypic variation in aggressiveness is indicative of pathogen adaptation [9,38]. In the case of *Cps*, such a phenotypic variation is evident. Aggressiveness varied among the isolates of different host origins. In the serial passage experiments, all boxwood isolates were consistent in aggressiveness, while the non-boxwood isolates changed dramatically during passages, suggesting that boxwood isolates have been well adapted, and the non-boxwood isolates were still in the process of evolution in this garden at the time of isolate collection. Comparing pachysandra and sweet box isolates, the *Cps* isolate of sweet box origin appears to adapt faster, as it resumed aggressiveness in the later passages after showing reduced aggressiveness in earlier passages, a trait not seen with the pachysandra isolate. If we assume that the sweet box isolate, sb3-P0, existed at the first report of boxwood infection in August 2015 in that garden, by the time of its collection two months later, it might have passed through the plant (completed its life cycle) four times. According to our experimentation, the total number of passages needed for *Cps* to become well adapted to sweet box is about 9 to 10 times. For pachysandra isolates, it appears that the same level of adaptation may take much longer: disease symptoms on pachysandra were found about 9 months after the initial observation of boxwood infection in the garden, and the isolate ph12-P0 did not show an ability to develop strong aggressiveness by the end of the passage experiment. The faster adaptation of *Cps* to sweet box raises an important question. How easily may *Cps* adapt to less-susceptible cultivars of *Buxus sempervirens* and other *Buxus* species, considering that the genetic differences are smaller between *B. sempervirens* ‘Suffruticosa’ and other *Buxus* species or cultivars than among the three host genera studied here? Alarmingly, the sweet box isolate sb3-P10 was as capable as its original sb3-P0, as was the boxwood isolate bw1-P10 in comparison to bw1-P0. This may suggest that the evolutionary pace for this fungus to adapt to *Buxus* with more resistance would not be slower than to adapt to sweet box.

This study once again demonstrated, using non-boxwood isolates, that adaptation to a novel host leads to reduced infectivity against former ones, causing attenuation, as has been shown previously [39]. Specifically, boxwood leaves inoculated with ph12-P10 and sb3-P5 developed significantly less infection than those inoculated with boxwood isolates bw1-P0, -P5, and -P10 (Figure 5a). Therefore, diversifying boxwood gardens by introducing more resistant boxwood varieties and non-host plants may be an effective means of boxwood blight control depending on how quickly the pathogen can adapt to these plants under local landscape conditions. Factors affecting the pace of adaptation should be considered when selecting and introducing garden plants. Sweet box is morphologically more like boxwood than is pachysandra; it is unclear whether this similarity in plant architecture or leaf physical characteristics might have some effect on the pace at which *Cps* isolates adapt to these host plants. Phylogenetically, pachysandra and sweet box are very close [40]. However, phenotypically, both boxwood and sweet box are woody, with smooth and shiny leaves arranged in a herringbone form, while pachysandra has larger, less-waxy leaves and an upright vining form [41,42]. Pachysandra was the last plant found to be infected by *Cps* in the garden where isolates were collected, and its *Cps* isolates showed more aggressiveness attenuation than those of sweet box origin. Thus, pachysandra might be a better companion plant than sweet box for boxwood in gardens for boxwood blight management purposes.

In addition to adaptation, host-associated specificity was also shown by the variation in the aggressiveness traits among the *Cps* isolates evaluated in this study. Traits important to infection and virulence are subject to rapid evolutionary change in a plant–pathogen interaction [43]. For non-boxwood isolates, all three aggressiveness components measured in this study seem to be important, as they varied dramatically during the passages. For boxwood isolates, sporulation may be particularly important for aggressiveness because it was the only trait that varied during the passage. However, the importance of aggressiveness traits may be affected by tradeoffs that can hamper selection of greater aggressiveness as described for many other pathogens [44,45,46,47]. Tradeoffs between aggressiveness traits of the isolates might have occurred in *Cps* in this study. We observed a significant tradeoff between lesion size and sporulation in both boxwood isolates and non-boxwood isolates. For boxwood isolates, even though they have limited lesion size due to small leaves, they produce large numbers of conidia before and after defoliation [5]. However, such a tradeoff is unlikely a negative factor for *Cps* aggressiveness or host adaptation since producing a large numbers of spores represents an alternative reproductive strategy that can make the evolution of mechanisms for host and mate choice unnecessary because it could lead to pathogen population persistence, transmission capacity enhancement and large mutational input on newer host plants [21]. In contrast, the tradeoff for pachysandra isolates that produced larger lesions but relatively low conidia per cm^2^ may be a negative factor affecting aggressiveness selection and epidemiological dynamics.

## Figures and Tables

**Figure 1 jof-09-00181-f001:**
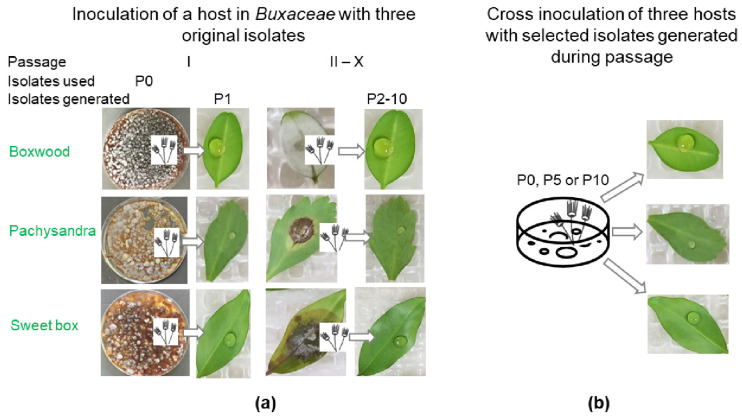
Schematic of the experimental setup. (**a**) Serial passage experiments on isolates’ originating host plants: boxwood, pachysandra, and sweet box, which generated isolates P1–P10. For passage I, inoculum was prepared from conidia formed on culture media (P0). For passages II–X, inocula were conidia from infected leaves of the previous passage (I–IX). (**b**) Post-passage experiments evaluating infectivity, lesion size, and ability to sporulate with selected original isolates (P0) and isolates generated during passages (P5 and P10) on all three host plants.

**Figure 2 jof-09-00181-f002:**
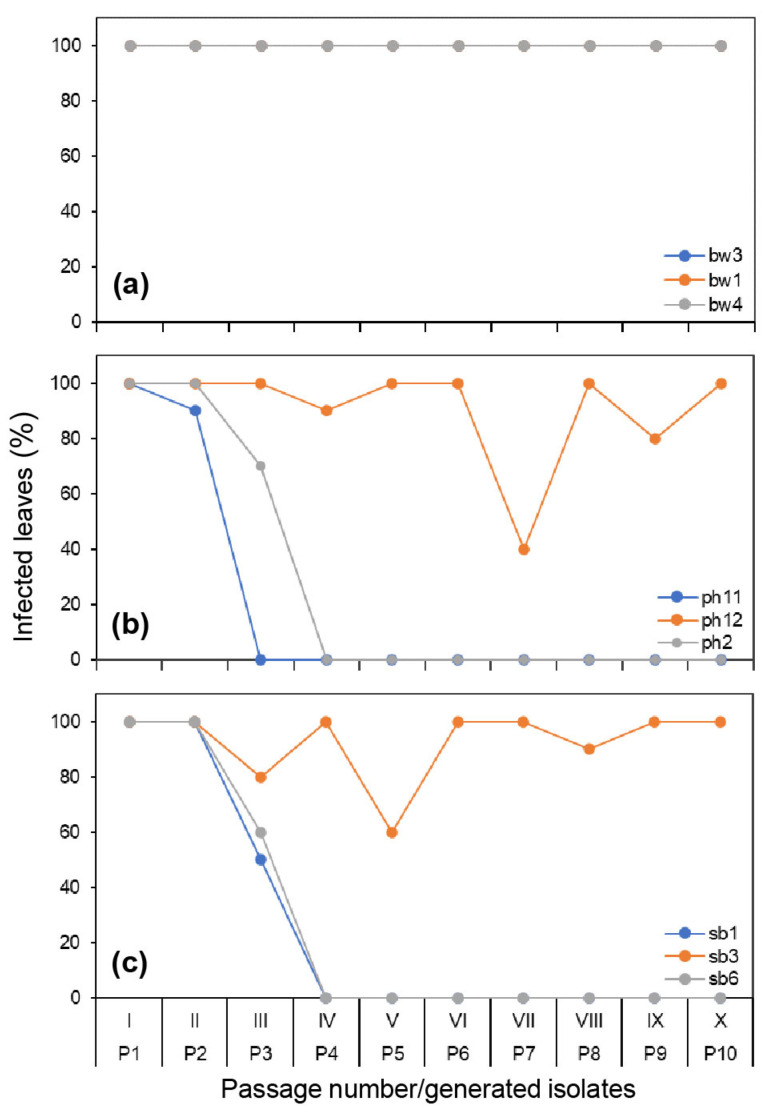
Infectivity of *Calonectria pseudonaviculata* isolates on their plants of origin, i.e., boxwood (**a**), pachysandra (**b**), or sweet box (**c**), 2 weeks after inoculation in 10 passages (I–X). The lines and dots of bw1 and bw3 are masked due to overlapping with bw4 in (**a**). For each isolate, detached leaves (*n* = 10) of an original host plant were inoculated with a drop of conidial suspension from a culture plate (P0) for passage I or from infected leaves of the previous passage I to IX for passages II–X. P1–P10 are the isolates generated in the individual passages. Each point represents an average of the infected leaves.

**Figure 3 jof-09-00181-f003:**
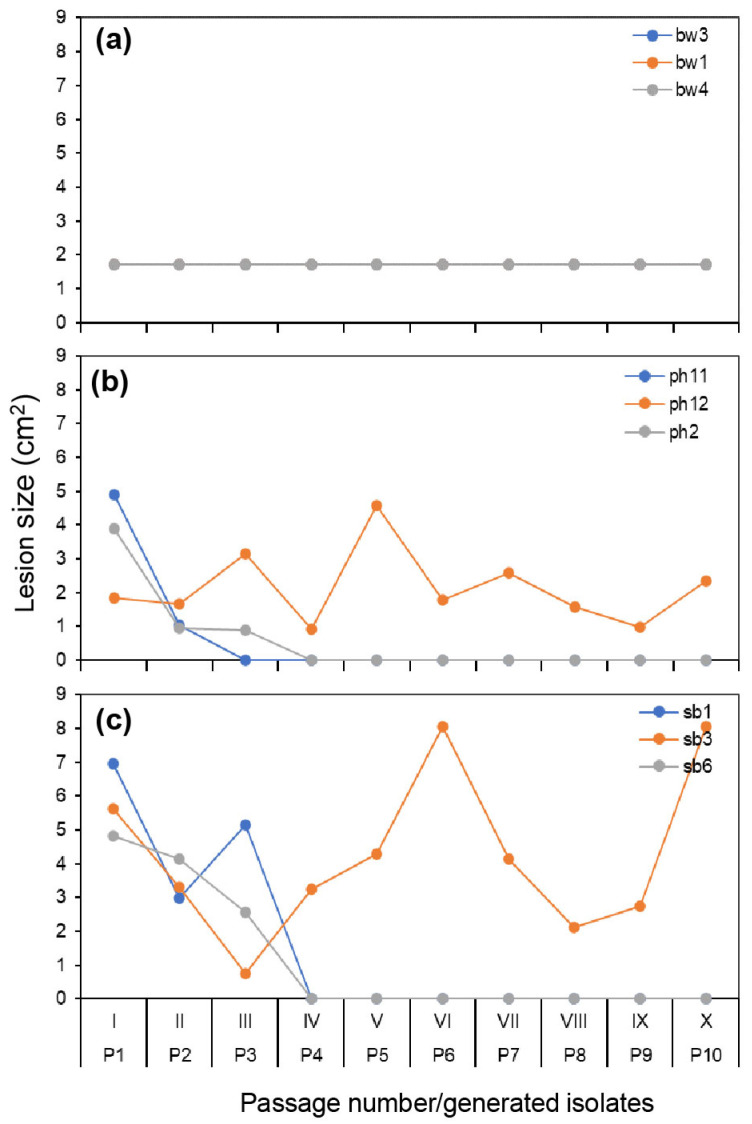
Blight lesion size caused by *Calonectria pseudonaviculata* isolates on their plants of origin, i.e., boxwood (**a**), pachysandra (**b**), or sweet box (**c**), 4 weeks after inoculation in 10 passages (I–X). The lines and dots of bw1 and bw3 are masked due to overlapping with bw4 in (**a**). For each isolate, detached leaves (*n* = 10) of an original host plant were inoculated with a drop of conidial suspension from a culture plate (P0) for passage I or from infected leaves of the previous passage I to IX for passages II–X. P1–P10 are the isolates generated in the individual passages. Each point represents an average of lesion sizes.

**Figure 4 jof-09-00181-f004:**
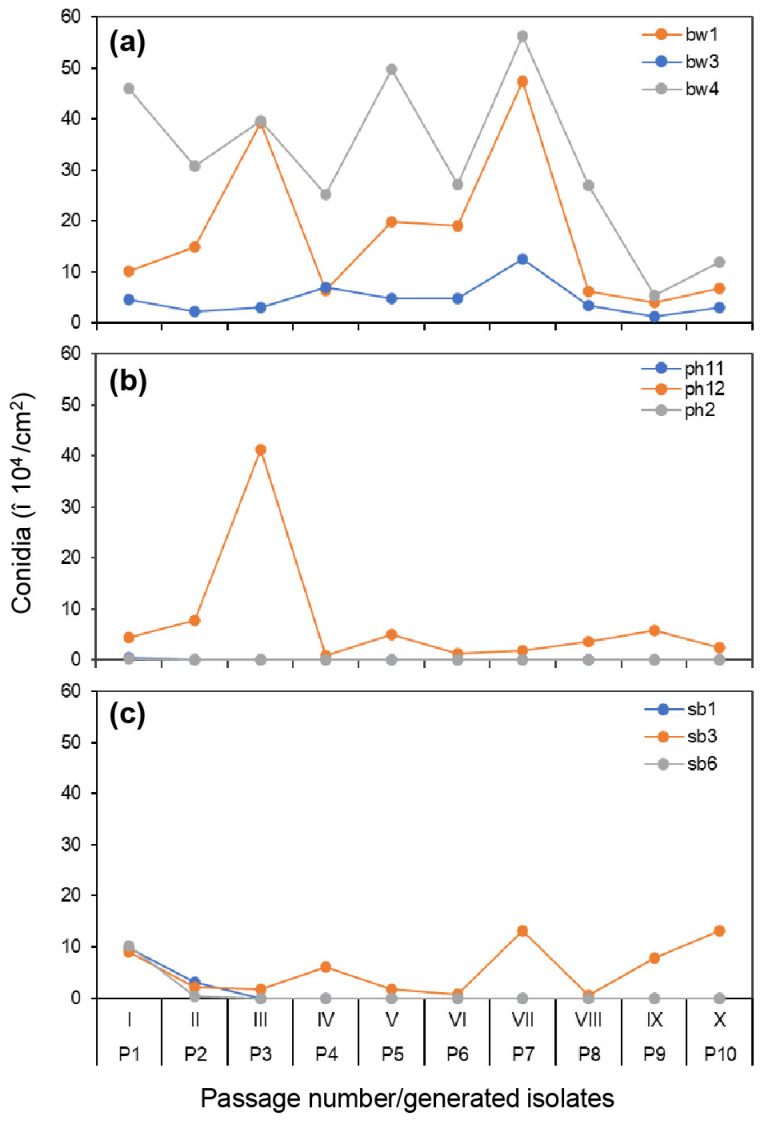
Conidia produced by *Calonectria pseudonaviculata* isolates on their plants of origin, i.e., boxwood (**a**), pachysandra (**b**), or sweet box (**c**), 4 weeks after inoculation in 10 passages (I–X). For each isolate, 10 detached leaves of an original host plant were inoculated with a drop of conidial suspension from a culture plate (P0) for passage I or from infected leaves of the previous passage I to IX for passages II–X. P1–P10 are the isolates generated in the individual passages. Each point represents an average of six counts (*n* = 6) of conidia collected from the infected leaves.

**Figure 5 jof-09-00181-f005:**
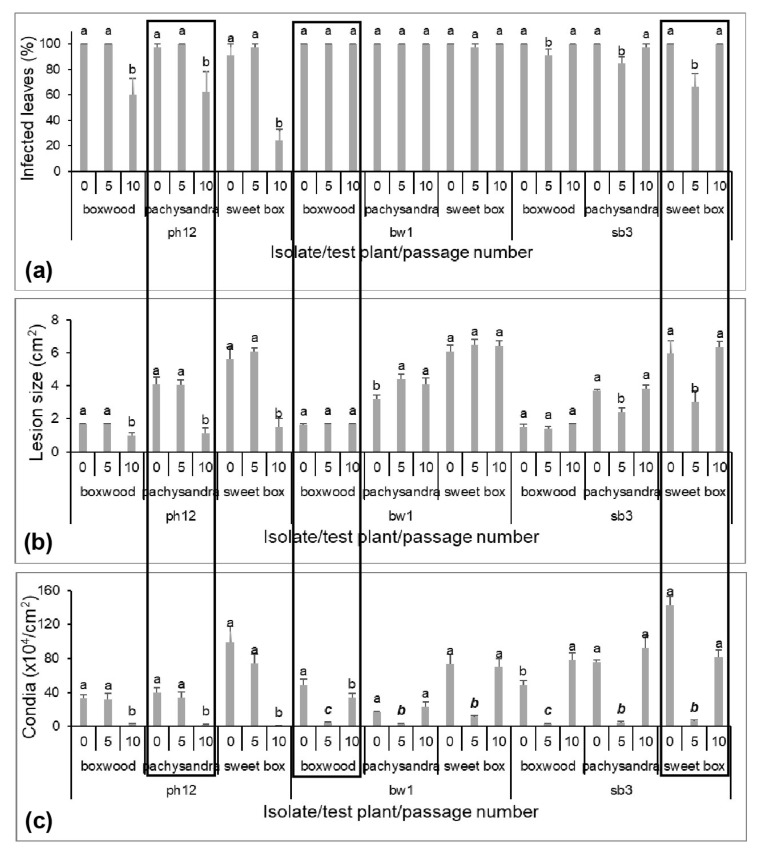
Comparison of aggressiveness components of selected *Calonectria pseudonaviculata* isolates before (P0) and after passages (P5 and P10) on different hosts. Infected leaves (**a**), lesion size (**b**), and conidial production (**c**) were assessed 14, 28, and 10 days after inoculation, respectively. Each column is an average of thirty leaves from three experiments. Columns are topped by a standard error bar and significance letter. For each isolate, columns from a single test plant topped with different letters are significantly different, *p* ≤ 0.05. Columns within broken-line rectangles depict aggressiveness of isolates on their original hosts.

**Table 1 jof-09-00181-t001:** Isolates of *Calonectria pseudonaviculata* used and generated in this study.

Isolate Name [15]	Passage Isolate Name ^X^	Host Plant	Date of Original Isolation
bw1	*bw1-P0*	*Buxus sempervirens* ‘Suffruticosa’	September 2016
bw3	bw3-P0	*B. sempervirens* ‘Suffruticosa’	
bw4	bw4-P0	*B. sempervirens* ‘Suffruticosa’	
p2	ph2-P0	*Pachysandra terminalis*	May 2016
p11	ph11-P0	*P. terminalis*	
p12	*ph12-P0*	*P. terminalis*	
sb1	sb1-P0	*Sarcococca hookeriana* var. *humilis*	October 2015
sb3	*sb3-P0*	*S. hookeriana* var. *humilis*	October 2015
sb6	sb6-P0	*S. hookeriana* var. *humilis*	November 2016
-	**bw1-P5**	*B. sempervirens* ‘Suffruticosa’	18 December 2017
-	**bw1-P10**	*B. sempervirens* ‘Suffruticosa’	15 March 2018
-	**ph12-P5**	*P. terminalis*	18 December 2017
-	**ph12-P10**	*P. terminalis*	16 March 2018
-	**sb3-P5**	*S. hookeriana* var. *humilis*	18 December 2017
-	**sb3-P10**	*S. hookeriana* var. *humilis*	26 March 2018

[15] See reference. ^x^ Isolate source indicated by bw (boxwood), ph (pachysandra), and sb (sweet box); isolate passage number is indicated by a “P” followed by the number after a hyphen; the original isolates that successfully went through 10 passages and were selected for further evaluation are shown in italics; isolates generated in this study are bolded.

**Table 2 jof-09-00181-t002:** Comparison of differential aggressiveness components during 10 passages of three *Calonectria pseudonaviculata* isolates on their plants of origin.

Variation Source	*Buxus* (Boxwood)	*Pachysandra* (Pachysandra)	*Sarcococca* (Sweet Box)	*p*-Value Among Hosts
	(*p*-Value Among Isolates)			
Infection rate (%)	100 (1)	45.7 (0.0002)	48.0 (0.0002)	0.0048
Lesion size (cm^2^)	1.7 (1)	1.1 (0.0171)	2.3 (0.0120)	0.4612
Conidia (×10 k/cm^2^)	18.0 (0.0003)	2.5 (0.0191)	2.6 (0.0104)	<0.0001
Infection cycle (days)	12.9 (1)	12.9 (1)	15.8 (1)	0.1034

**Table 3 jof-09-00181-t003:** Post-passage analysis of significance of factors affecting aggressiveness components of the selected *Calonectria pseudonaviculata* isolates from passages 0, V, and X on three hosts.

Variation Source	Infection (%)	Lesion Size (cm^2^)	Conidia (10^4^/cm^2^)
Passage	<0.0001	<0.0001	<0.0001
Isolate	<0.0001	0.0010	<0.0001
Host	0.0005	<0.0001	<0.0001
Experiment	0.0174	0.7007	0.3627
Passage × Isolate	<0.0001	<0.0001	<0.0001
Passage × Host	0.4008	0.1618	<0.0001
Passage × Experiment	<0.0001	0.1352	0.1644

## Data Availability

Not applicable.

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
