# Peer review of "Differential Adaptation Has Resulted in Aggressiveness Variation of *Calonectria pseudonaviculata* on Hosts *Buxus*, *Pachysandra*, and *Sarcococca"

_jof, 2023, doi:10.3390/jof9020181_

Round 1
Reviewer 1 Report
Manuscript entitled “Differential adaptation has resulted in aggressiveness variation
of Calonectria pseudonaviculata on hosts Buxus, Pachysandra and Sarcococca”. The manuscript investigated the adaptation of Calonectria pseudonaviculata to three hosts. The results had significance for studying aggressiveness variation of Calonectria pseudonaviculata. However, several points need to be addressed before it can be published.
1. Why the author chose Buxus, Pachysandra and Sarcococca three hosts for investigate the aggressiveness variation of Calonectria pseudonaviculata? What the relationship among the three hosts? Are the three hosts had representativeness?
2. The evolution of microbe normally undergo long period. Why the author chose 10 passages for adaptation experiment, but not more passages? Are there any reference supported?
3. The most important, the results showed that difference was exists among the three hosts. Therefore, in the Discussion section, the author must deeply discussed why the difference was exists among the three hosts? Are these difference caused by plant-microbe interaction or other mechanisms?
Author Response
- Why the author chose Buxus, Pachysandra and Sarcococca three hosts for investigate the aggressiveness variation of Calonectria pseudonaviculata? What the relationship among the three hosts? Are the three hosts had representativeness?
Calonectria pseudonaviculata (Cps) is relatively conserved genetically but its aggressiveness variation has been described among Buxus, Pachysandra and Sarcococca, which are the three identified host genera, all belonging to Buxaceae. Cps infects multiple species of all three of these genera in spite of their morphological and genetic differences. The plants we chose to work with are popular species/cultivars for landscape planting, representative of Buxaceae used in gardens.
- The evolution of microbe normally undergo long period. Why the author chose 10 passages for adaptation experiment, but not more passages? Are there any reference supported?
Serial-passage experiments are a form of experimental evolution used to study pathogen adaptation and co-evolution between host and pathogen. The number of passages is dependent on how soon the pattern of interaction changes between the pathogen and its hosts. We added one more reference and also listed related works in the introduction in response to this comment. We chose 10 passages based on our preliminary experiment, which showed significant changes even before the 5th passage.
- The most important, the results showed that difference was exists among the three hosts. Therefore, in the Discussion section, the author must deeply discussed why the difference was exists among the three hosts? Are these difference caused by plant-microbe interaction or other mechanisms?
Yes, Cps aggressiveness differed among hosts as we showed in the previous and also this study. However, the mechanisms are not clear, which deserves further investigation that is beyond the scope of this study. We focused on the pathogen’s adaptation to different hosts—there were changes in pathogen aggressiveness over time, even across a single host.
Reviewer 2 Report
Please see attached file of comments

Author Response
The article is very interesting and refers to the most important issue which is the difference between the fitness and virulence of a pathogen which is very important to plant pathologists. I think that this work is considered novel in this area of research. Although of that, there are some comments mentioned below:
Abstract
It is clear
Thank you much for the comments.
Introduction
It is well written but it is important to refer to the taxonomic position of the pathogen
The taxonomic position of the pathogen is added.
Materials and Methods
Each subtitle of the Materials and Methods should be separated into one line and numbered
Revised.
Results
The results are clear and explained precisely
Thanks for the comments.
Discussion
It is well written, but there is no interpretation of why the sexual reproduction of the pathogen is
still unknown. The authors should explain this.
Added the explanation and reference.
Line 101: What is the name of bleach used
Added the name and company location.
The title of Table 1. Is confused and unclear.
Sorry about this. This was a mistake that the tile was not c/p.
The last six rows in the second column are not in bold.
They are in bold on my end, but I would leave it for the editors.
I think the word offspring is not suitable.
I changed it to descendant/ descendants.
Results
In Table 2. Cm2 should be cm2
(2 is superscript)
Revised.
Line 251: Calonectria pseudonaviculata should be in italic
Revised.
The following are some corrected English errors:
Line 9: (boxwood) Pachysandra (pachysandra) and correct to (boxwood), Pachysandra
(pachysandra), and
Corrected.
Line 14: previous inoculation correct to the previous inoculation
Corrected.
Line 14: The sentence starting with Boxwood must be revised and modified
Revised.
Line 20: most adapted correct to the most adapted
Corrected.
Line 21: coevolution correct to coevolutionary
Corrected.
Line 30: Calonectria pseudonaviculata correct to Calonectria pseudonaviculata (italic)
Corrected.
Line 31: The sentence starting with Boxwood blight must be revised and modified
Revised.
Line 33: of three correct to of the three
Corrected.
Line 34: Study of correct to A study of
Corrected.
Line 48: mycelial correct to the mycelial
Corrected.
Line 51: The sentence starting with The aggressiveness must be revised and modified
Revised.
Line 62: for emergence correct to for the emergence
Added “The”.
Line 63: host imposed correct to host-imposed
Corrected.
Line 71: on different correct to in different
Corrected.
Line 72: selection correct to the selection
Corrected.
Line 74: understand correct to understand the
Corrected.
Line 87: from passages and impact correct to in passages and the impact
Corrected.
Line 90: Calonectria pseudonaviculata should be in italic
Corrected.
Line 92: sweet box correct to the sweet box
NA
Line 138: passage correct to the passage
Corrected.
Line 139: P1 to P10 correct to P1-P10
Corrected.
Line 142: obtained correct to obtain
Corrected.
Line 143: conidial correct to conidia
Corrected.
Line 146: Comparative correct to A Comparative
NA
Line 154: passage I correct to the passage I
Corrected.
Line 160: The sentence starting with Specifically must be revised and modified
Revised.
Line 164: sweet box correct to the sweet box
NA
Line 169: for boxwood correct to for the boxwood
NA
Line 171: was analyzed correct to were analyzed
Corrected.
Line 174: Data of correct to Data from
Corrected.
Line 176: Means correct to The means
Corrected.
Line 180: with consistent correct to with a consistent
Corrected.
Line 181: and sweet box correct to and one sweet box
Corrected.
Line 193: The sentence starting with Specifically must be revised and modified
Revised.
Line 195: had stronger correct to had a stronger
Corrected.
Line 215: in the course correct to over the course
Corrected.
Line 219: The sentence starting with Looking at must be revised and modified
Revised.
Line 276: The sentence starting with Although Calonectria must be revised and modified
Revised
Line 280: differential correct to a differential
Corrected.
Line 282: sweet box correct to a sweet box
NA
Line 282: coevolution the coevolution
Corrected.
Line 283: appropriate correct to the appropriate
Corrected.
Line 284: management correct to the management
Corrected.
Line 289: significant correct to a significant
Corrected.
Line 293: The sentence starting with Obviously must be revised and modified
Revised.
Line 294: that may correct to may
Corrected.
Line 299: to occurrence correct to to the occurrence
Corrected.
Line 306: from accidental introduction correct to from the accidental introduction
Corrected.
Line 310: having occurred correct to having to occur
Revised.
Line 324: passaged on correct to passed through
Corrected.
Reviewer 3 Report
the research is interesting. it could also be accompanied by some mechanism (monitoring the level of specific enzymes or genes) that supports adaptability. at first glance it looks unfinished. adaptability is a very complex process that depends on both the pathogen and the host. I don't understand why only the leaves were used in the research and not the whole plant? monitoring the changes in the leaves that are physiologically functional would give a better result. it would be good if, in the introduction, the importance of such research in relation to monitoring the changes on the leaves of the plant during growth and whether the obtained results are valid enough, especially in the sense mentioned in the discussion about potential mechanisms of adaptation (eg line 293) .
line 336-353 discussion presents interesting possibilities of applicability of the results related to the use of resistant varieties, but they were not examined in this paper. the work would be much better if this boxwood group were added.
Author Response
the research is interesting. it could also be accompanied by some mechanism (monitoring the level of specific enzymes or genes) that supports adaptability. at first glance it looks unfinished. adaptability is a very complex process that depends on both the pathogen and the host.
This is a very good idea that will be included in our future study to understand the mechanisms of adaptation. This study focused on aggressiveness changes from adaptation.
I don't understand why only the leaves were used in the research and not the whole plant? monitoring the changes in the leaves that are physiologically functional would give a better result. it would be good if, in the introduction, the importance of such research in relation to monitoring the changes on the leaves of the plant during growth and whether the obtained results are valid enough, especially in the sense mentioned in the discussion about potential mechanisms of adaptation (eg line 293) .
Performing serial passage experiments with plants was not feasible and practical. First, boxwood and sweet box are woody plants that take years to grow to get decent sizes for experiments. Second, inocula levels for plants with different sizes are hard to control. Third, Cps is a quarantine pathogen in the experimenting area and must contained in the lab and we do not have Lab space or growth chamber to do the serial passage experiments that three hosts and their replicate plants and 10 passages. Also when plants get infected, leaves defoliated that is hard to track down the lesion sizes and sporulation. We have validated detached leaf inoculation in our previous study, which is easy, effective, and economic.
line 336-353 discussion presents interesting possibilities of applicability of the results related to the use of resistant varieties, but they were not examined in this paper. the work would be much better if this boxwood group were added.
Agree.
Round 2
Reviewer 1 Report
All the comments had been addressed.
Reviewer 3 Report
the authors answered all my questions and incorporated the main suggestions into the text